# Ex Vivo Visualization of Distribution of Intravitreal Injections in the Porcine Vitreous and Hydrogels Simulating the Vitreous

**DOI:** 10.3390/pharmaceutics15030786

**Published:** 2023-02-27

**Authors:** Tobias Auel, Lara Paula Scherke, Stefan Hadlich, Susan Mouchantat, Michael Grimm, Werner Weitschies, Anne Seidlitz

**Affiliations:** 1Institute of Pharmaceutics and Biopharmaceutics, Heinrich Heine University Düsseldorf, 40225 Düsseldorf, Germany; 2Center of Drug Absorption and Transport, Department of Biopharmaceutics and Pharmaceutical Technology, Institute of Pharmacy, University of Greifswald, 17489 Greifswald, Germany; 3Institute of Diagnostic Radiology and Neuroradiology, University Medicine Greifswald, Ferdinand-Sauerbruch-Straße, 17475 Greifswald, Germany

**Keywords:** magnetic resonance imaging, vitreous body, vitreous substitute, intravitreal drug delivery, hydrogel

## Abstract

The characterization of intravitreal dosage forms with regard to their behavior in vivo is usually explored in preclinical development through animal studies. In vitro vitreous substitutes (VS) to simulate the vitreous body for preclinical investigations have so far been insufficiently studied. To determine a distribution or concentration in the mostly gel-like VS, extraction of the gels is required in many cases. This destroys the gels, which makes a continuous investigation of the distribution impossible. In this work, the distribution of a contrast agent in hyaluronic acid agar gels and polyacrylamide gels was studied by magnetic resonance imaging and compared with the distribution in ex vivo porcine vitreous. The porcine vitreous served as a surrogate for human vitreous since both are similar in their physicochemical properties. It was shown that both gels do not completely represent the porcine vitreous body, but the distribution in the polyacrylamide gel is similar to that in the porcine vitreous body. In contrast, the distribution throughout the hyaluronic acid agar gel is much faster. It was also shown that anatomical features such as the lens and the interfacial tension to the anterior eye chamber could have an influence on the distribution that is difficult to reproduce using in vitro VS. However, with the presented method, new in vitro VS can be investigated continuously without destruction in the future, and thus their suitability as a substitute for the human vitreous can be verified.

## 1. Introduction

Intravitreal injection of drugs is the most commonly used method for treating diseases of the posterior segment of the eye, such as diabetic retinopathy or age-related macular degeneration [1,2]. In this process, an active ingredient is injected into the human vitreous body as a solution, suspension, or implant. With a volume of 4.0–4.5 mL, the vitreous body represents the largest part of the human eye and consists of 98–99% water, electrolytes, and proteins. Type II collagen and glycosaminoglycan hyaluronic acid give the vitreous body a gel-like structure [3,4,5]. The site of action of intravitreally injected drugs is, in many cases, the retina. The onset and duration of the therapeutic effect, therefore, depends on how it is distributed throughout the vitreous and reaches the target tissue. The distribution of the injected active ingredient in this framework is controlled by diffusion and convection processes [6,7,8].

According to the Stokes–Einstein equation and the Hagen–Poiseuille law, both diffusion and convection depend on the viscosity, which is why this takes an important role in the distribution of the vitreous body [8]. However, the viscosity of the human vitreous also depends on its age. The age-related liquefaction of the vitreous body due to an accumulation of collagen fibrils causes a reduction in viscosity [9,10,11]. Since this is not linear, it makes it difficult to accurately predict the distribution in the human vitreous body.

Animal models are often used to study the release and distribution of new drugs in preclinical development. Pigs, rabbits, mice, and monkeys are typically part of intravitreal studies [12,13,14,15,16]. In addition to the ethical aspects of animal studies, the transferability of the results is problematic in many cases. According to Laude et al., different volumes of the vitreous bodies and, thus, also different sizes result in different diffusion paths [17]. Furthermore, the composition of the vitreous body and the elimination pathways vary. This, in combination with the need to reduce the number of animal trials and the fact that only the most promising formulations in those studies are used, leads to a need for in vitro models to study distribution and drug release. Using standardized models such as the flow-through cell (USP Dissolution Apparatus 4) or the reciprocating holder (USP Dissolution Apparatus 7) may provide reproducible results, but the conditions prevailing in these setups typically have little in common with the physiological or pathophysiological conditions at the actual site of application. The limitations include the different transport mechanisms in stirred dissolution media as opposed to the gelled and unstirred vitreous and the volumes used for the studies [18,19]. Therefore, more biorelevant models, such as the PK-Eye by Awwad et al. or the vitreous model by Loch et al., have already been developed to address different biopharmaceutical aspects [20,21,22]. Both models are based on hydrogels used as in vitro vitreous substitutes (VS). The PK-Eye was developed as a model to simulate the clearance of a drug through the anterior chamber of the eye and uses a hyaluronic acid agar gel as a vitreous substitute. The Vitreous model by Loch et al. was designed to investigate the distribution of a drug in a polyacrylamide (PAA) gel. In initial studies, Loch et al. investigated the distribution of the model drug fluorescein sodium under different influences [23,24]. All these studies have in common that for visualization of the distribution or determination of a concentration in the VS, the hydrogels have to be extracted. This leads to an experimental design with multiple endpoints and does not allow continuous tracking of the distribution of the drug.

The aim of this work was to develop a technique to continuously observe the distribution of model drugs in in vitro VS without destroying them and thus breaking off the experiment. Using this method, it should be possible to compare different VS with each other and with ex vivo vitreous bodies to find a suitable in vitro VS. This could ideally lead to a reduction of necessary animals in preclinical studies of new intravitreal dosage forms. For this purpose, VS already described in the literature were used, and the distribution of contrast agents in them was investigated by using magnetic resonance imaging (MRI). The distribution was additionally studied in ex vivo experiments in porcine eyes. The porcine vitreous, in many aspects, approximates the human one and thus serves as a surrogate for it [25].

## 2. Materials and Methods

### 2.1. Artificial Vitreous Substitutes

Phosphate buffered saline (PBS) pH 7.4 consisted of sodium chloride, potassium chloride, disodium hydrogen phosphate, and potassium dihydrogen phosphate (all AppliChem, Darmstadt, Germany). For the preparation of the 2% PAA gel, the individual components were added one after the other volumetrically to a glass beaker and mixed with a glass rod, and gel formation took place within 30 min. Reduction of ammonium persulfate (Carl Roth GmbH, Karlsruhe, Germany) by adding TEMED (Carl Roth GmbH, Karlsruhe, Germany) results in radical formation and polymerization of the acrylamide monomers. The bisacrylamide (Rotiphorese^®^, Carl Roth GmbH, Karlsruhe Germany) provides crosslinking of the otherwise linear framework, resulting in a gel structure. To prepare the 0.15% HA gel, hyaluronic acid (MW 350 kDa) and agar (both Caesar & Loretz GmbH, Hilden, Germany) were each suspended separately in PBS 7.4 and boiled until a clear solution was obtained. Subsequently, both solutions were mixed and cold-stirred to room temperature. The composition of the individual hydrogels is listed in Table 1. Compared to the original HA gel, the concentration of the gel-forming components was reduced by 25% each to adjust the viscosity.

### 2.2. Ex Vivo Material

Pig eyes were used to compare the artificial VS with ex vivo data. These came as by-products from a local slaughter and were processed within two hours after slaughter. During this time, the eyes were stored in PBS 7.4 to prevent desiccation. Surrounding skin layers were removed, and remnants of the optic nerve were separated. Some of the porcine eyes were subsequently frozen at −20 °C to study the effect of storage in a deep freezer on the vitreous. 

### 2.3. Physicochemical Characterisation

The pH of the VS was measured using a FiveEasy FE20 pH electrode (Mettler Toledo, Gießen, Germany). A Densito density meter (Mettler Toledo, Gießen, Germany) was used to determine the density of the VS. Due to the risk of contamination with animal material, determining the density of the animal vitreous was performed using a pycnometer because it could be thoroughly cleaned and disinfected. A Brookfield DV-II+ rotational viscometer (AMETEK GmbH, Berwyn, PA, USA) was used for the viscosity measurements. The measurements were carried out at 20 °C and 20 rpm. All experiments were performed after complete gelation, in triplicate, and at room temperature unless otherwise indicated. In each case, porcine eyes were opened with tissue scissors immediately before measurement so the vitreous could be obtained for characterization.

### 2.4. MRI Measurements

MRI scans were performed in a 7.1 Tesla MRI scanner (ClinScan 70/30, Bruker Bio-Scan, Billerica, MA, USA) with a receive-only 2 × 2 rat brain surface coil (Bruker, Billerica, MA, USA). A T1-weighted turbo spin coronal echo sequence imaging was performed. The individual parameters are listed in Table 2. The vitreous substitute was positioned directly under the coil using a 3D-printed sled (see Figure 1). A 3D-printed sled and the attachment on it allowed the gel-like VS to be held in shape and positioned accurately. The sled itself was printed of Polylactide filaments (Formfutura, Nijmegen, The Netherlends) with an Ultimaker 3 (Ultimaker, Utrecht, The Netherlands), and the mold for the VS was made with a Form 3 printer using clear Resin (both Formlabs, Somerville, MA, USA). The individual models were designed using FreeCAD (version 0.17) and sliced using either Preform (version 3.2.0) or Cura (version 4.4.0). Besides a central cavity containing the vitreous substitute or porcine vitreous, respectively, there were 4 hollow mantle segments for reference standards included in the setup. 

A native gel and gels mixed with a contrast agent were filled into a hollow mantle segment surrounding the VS for the artificial VS. The containers with contrast agents were not considered for evaluation in this work. A native measurement was performed before the start of each imaging session. Afterward, the vitreous to be examined was moved out of the MRI, and 0.1 mL of a 0.01 mmol/mL gadobutrol solution (GadoVist^®^, Bayer, Leverkusen, Germany) was injected. Injection was performed using Injekt^®^-F Solo 1 mL fine-dose syringe (B. Braun, Melsungen, Germany) and a cannula (Sterican 0.4 × 40 mm). To ensure a reproducible injection, 3D-printed distance holders for the cannula were used. Measurements were run every 60 min for a total duration of 12 h; samples were stored at room temperature between the measurements. 

### 2.5. MRI Image Processing

The MRI images were evaluated using the open-source software HOROS (version 4.0 RC5, Horos Project). The injected contrast agent increased the signal intensity, which was used to evaluate the gel volume in which the contrast agent had distributed. The evaluation for the artificial VS was performed against a native VS filled into the external hollow mantle segments as a reference. The maximum value of this reference was determined and multiplied by 1.25; all values above this were defined as gel portions containing contrast medium. Due to the co-traveling reference within the slices, this could be determined again for each layer, so coil distance effects were reduced. Of the 40 slices, the first and last five slices were each excluded because they could not be evaluated for all VS. Using HOROS, a 3D model was generated of the coronal layers by internal software tool from the contrast agent surfaces of the individual slices, from which the contrast agent-added volume was calculated. Figure 2 shows the course of the individual steps of the evaluation. 

In the evaluation of the porcine eyes, no standard was co-measured due to the individual differences of each eye. Here, the native image for each slice was taken as a reference. The maximum value multiplied by a factor of 1.25 was also set as the contrast agent limit. The evaluation of the individual slices was then performed analogously to the artificial VS (see Figure 3).

## 3. Results

The physicochemical parameters shown in Table 3 were determined to compare the VS with the ex vivo data. These were also compared with human vitreous data from the literature. The target range of pH should be 7.4. As expected, this was also determined for the porcine vitreous body. The produced VS deviated minimally from this. When determining density, the porcine vitreous is slightly below the human literature data, and a large variability is also visible. The VSs have a higher density than the porcine vitreous, with the HA gel matching the human vitreous literature data. Viscosity is difficult to compare with literature data because the conditions under which they were collected are either not specified in detail or cannot be replicated. These data have only been given for the sake of completeness. The measured viscosity of the porcine vitreous is firmly below the literature values for the human vitreous. The HA gel is approximately in the porcine vitreous range, while the PAA gel values reach 10-fold higher viscosities.

The distribution of the contrast agent GadoVist^®^ was investigated in the VS and animal porcine eyes. In each measurement, 40 transverse slices were recorded from top to bottom. To assess the effect of frozen storage on vitreous, pig eyes were divided into two groups: first, fresh pig eyes that were processed immediately after slaughter were examined, and second, those that were frozen for at least 24 h after slaughter and thawed before the examination.

Figure 4 shows the distribution of the contrast agent over time in different areas of a measured “fresh” porcine vitreous body. Here, the distribution in layer 10 above the lens, in layer 20 in the center of the vitreous, and in layer 30 below the lens are shown as examples. With increasing distance from the coil, the image becomes darker due to reduced absolute signal intensity, but contrast ratios remained comparable. It can be seen that initially the largest amount of contrast agent can be recovered at the central injection site (layer 20). In the upper region (layer 10), a small amount of contrast agent is present due to the injection process, while no contrast agent is yet detectable in the lower regions (layer 30). This is followed by a distribution into all areas over time, which is almost uniform throughout the entire vitreous after about six hours.

Figure 5 shows the MRI images of the distribution of the contrast agent over time in a vertical cross-section. This cross-section in the frontal direction onto the VS, or porcine vitreous, was generated from the measured coronal slices discussed above as a composite image by the HOROS software. In the PAA gel, an injection cloud concentrated in the center can be seen, which spreads uniformly in all directions over the investigation time. In contrast, the initial injection spot in the HA gel is larger following the injection angle and stretches from the center to the edge of the VS. Here, too, a uniform distribution can be seen. After four hours, only the outer areas are free of contrast agents. After six hours, the distribution in both VS is almost complete. In the porcine vitreous, a strong adherence of the contrast agent to the lens is recognizable after injection. Starting from the lens, a distribution into posterior sections of the vitreous occurs. The difference between thawed and fresh vitreous is striking. While in the freshly measured vitreous body, a separation between the vitreous body and the anterior chamber of the eye is still recognizable, this no longer appears to be present in the thawed vitreous body. During the injection into the thawed porcine vitreous body, a distribution of the contrast medium into the vitreous body and into the anterior chamber of the eye can be seen. After 6 h, the contrast medium is almost completely distributed in both porcine vitreous bodies.

The percentage volume in which the contrast agent could be detected is shown in Figure 6. For the porcine vitreous, the eye’s anterior chamber was also considered a potential distribution volume. It can be seen that the injected GadoVist^®^ solution initially distributes differently. While this distribution throughout the entire volume is lowest directly after injection into the PAA gel with 10.8%, it is already 38.7% in the HA gel. With 16.2% (fresh) and 28.3% (thawed), the pig eyes are located between the two VS. After that, a rapid distribution occurs, which is fastest for the HA gel and the thawed pig eye. Thus, the contrast medium was detected in 90% of the volume for the HA gel and the thawed pig eye after only 4 h, whereas this value was reached later for the directly measured pig eye and PAA gel (7 and 9 hours, respectively). All test series reach a plateau after 10 h at the latest so that a complete distribution can be assumed.

## 4. Discussion

In ophthalmology, speaking of VS, it is most often referred to as in vivo substitutes. The human vitreous is responsible for maintaining the eyeball’s shape, allowing the circulation of nutrients and other solute molecules, contributing to the stability of intraocular pressure, and holding the lens and retina in place [28]. If it can no longer fulfill its function due to, for example, liquefaction, vitreous detachment with a retinal lesion or vitreous hemorrhage may occur. This is followed in many cases by vitrectomy, in which part of the vitreous is removed and replaced by a VS after treatment. These VS are used as a temporary or permanent tamponade to the retina to maintain pressure on it [27]. The search for the ideal VS in this context is a challenging, seemingly unsolvable task. The ideal VS should be clear and transparent, similar to the human vitreous body in terms of refractive index, viscosity, density, and osmolality, and meet other physical and biological requirements [27,28,29,30]. While the search for VS for in vivo application is a major component of current research, the simulation of the vitreous body in vitro has played a minor role. The requirements for an ideal in vivo VS are not directly transferable for in vitro application because factors such as the transmission of light, toxicological aspects, or intraocular pressure are of secondary importance for drug release in vitro. Current VS used in vivo, such as silicone oils, perfluorinated alkanes, or gases, are unsuitable for in vitro application in release testing which should represent the non-vitrectomized eye due to their nature.

Therefore, a literature search identified several gels that could be used as VS. Two of these were prepared and used in this work. One is the PAA gel published by Loch et al. [22], and the other is the HA gel used in several sources [20,31]. While the former consists of exogenous components, hyaluronic acid, as the main component of HA gel, is also one of the essential scaffold builders in the vitreous. The second principal component of human vitreous, collagen, on the other hand, is not present in any of the used gels. This could be due to the necessary quality of the biological material. While hyaluronic acid can be produced biotechnologically using yeast, collagen extraction as an animal product is associated with a high effort. On the one hand, this results in high costs and, on the other hand, in higher quality variability. For this reason, the investigation of gels with added collagen was not carried out in this work.

The results of the characterization of VS were compared with literature data on the human vitreous body. The aim was to get as close as possible to the human vitreous body in terms of pH value, density, and viscosity. Since the characterization of human vitreous bodies proved difficult for ethical and practical reasons, additional ex vivo taken porcine vitreous bodies, which resemble the human vitreous body in essential properties, were characterized. In this present work, all gels used as in vitro VS could be more or less adjusted to the desired pH of 7.4 by using PBS. This is relevant in that many active ingredients have pH-dependent properties. A change in the pH value can, for example, change the solubility, affecting the release or distribution. A fluctuation of the pH value around 7.4 could also be observed in the human vitreous body. Stein et al. found pH values of 7.54 ± 0.34 in a study with post-mortem human vitreous [32]. Next to the pH value, density is an important parameter when it comes to intravitreal formulations. According to Stokes’ law, density plays a role in the speed of sedimentation of particles. Thus, variations in density could be relevant when investigating the behavior of intravitreal suspensions. For example, it was shown by Stein et al. that an injected triamcinolone acetonide suspension in PAA gel remains at the injection site, whereas it sediments in a liquefied VS [24]. Both hydrogels and porcine vitreous examined were similar to literature values of human vitreous. In the results in Table 3 only a small deviation of the density values can be seen. As similar as the hydrogels are in pH and density, they are different in viscosity. According to the Stokes–Einstein equation, viscosity has an influence on diffusion processes, as mentioned before. The viscosity of the PAA gel is ten times higher than that of the HA gel. When trying to compare the gels with literature data for the human vitreous, it is noticeable that the comparison is difficult due to the data collection of the ex vivo values. Often, it is not comprehensible how the values were collected. Even the use of a different device, temperature, or method can lead to different results. The age of the vitreous bodies used for rheological examinations also varies. In most cases, older donors are involved, in which age-related liquefaction is already well advanced. Data on young human vitreous are currently not described in the literature. Based on the work of Sebag et al. from 1987, in which the age-related changes of the gel structure are discussed, it can be assumed that a simple estimation of the viscosity of the human vitreous body is not possible [11]. Nevertheless, in order to have a target point for viscosity, a determination of the viscosity of porcine vitreous bodies was performed in this work since these have a high structural similarity to human vitreous bodies [25,33,34]. Comparing the viscosities, the PAA gel seems to have a surprisingly high viscosity, while the HA Gel has similar values compared to the porcine vitreous. However, it must be noted that mechanical impairment of the porcine vitreous body cannot be ruled out due to the extraction process. During the viscosity measurement, a rheodestruction of the vitreous body could also be detected. Based on the results of the characterization of the hydrogels, it would be expected that for the distribution of active ingredients, the HA gel seems to be a more suitable model because of its similarities to the porcine vitreous in viscosity.

The distribution inside hydrogels as VS has already been investigated with model substances [23,24]. The continuous tracking of the substances was a major challenge. A quantitative determination of the distribution requires a concentration determination in which the hydrogels are separated into individual parts, extracted and destroyed in the process. This leads to experimental designs with many experiments and different endpoints, whereby a continuous tracking of the distribution cannot be carried out. This problem was avoided by using MRI. The use of ex vivo material limits the time over which distribution processes can be measured. Prolonged storage leads to drying out here and, thus, to a change in shape and volume. Therefore, a solution of a quickly distributing contrast agent was chosen for this work. At this point, it must be mentioned that the visualization of therapeutically used active substances is currently not possible with this method.

The 3D-printed device allowed placement of the VS directly under the coil in combination with exact repositioning for the next sampling time. The injection volume was set to 100 µL, which corresponds to a commonly therapeutically injected volume of intravitreal injection in vivo [35]. The PAA gel and the HA gel were used as artificial VS to investigate the influence of viscosity and gel former. In addition, pig eyes obtained from a local butcher were studied as an ex vivo comparison. Since a measurement cycle lasted twelve hours, one eye was measured “fresh” promptly after collection, while the second was immediately frozen and thawed before measurement. This was done to avoid wasting biological material and determine the effect of frozen storage on the porcine vitreous. GadoVist^®^ as a model drug was distributed very rapidly in the HA gel. Immediately after injection, the contrast agent was found in just under 40% of the VS, and after two hours, in almost 80%. In the PAA gel, the contrast medium is initially found in just under 18% of the volume after injection, and 80% is reached only after 6 h.

The images after the injection in Figure 5 shows an initially different injection spot. While in the higher viscosity PAA gel, a spherical injection cloud limited to the center is visible, in the HA gel, distribution into the outer gel is visible, most likely due to the lower viscosity. The injection cloud in the porcine vitreous seems to be more similar to that in the PAA gels, although a much higher viscosity was measured in the latter. Thus, it could be that viscosity plays only a minor role in the distribution of substances in hydrogels and that other aspects, such as the nature of the gel-forming components, are of more crucial importance, or the in vitro viscosity measurement of vitreous bodies is not representative due to the necessary extraction and the associated mechanical stress. In the MRI images of the porcine vitreous, an accumulation of the contrast agent in the posterior region of the lens is also visible. Due to the central injection, the lens might initially serve as a barrier for diffusion into the anterior chamber of the eye, which is why the contrast agent seems to accumulate around it. The relevance of the position of an intravitreal injection was described by Friedrich et al. in a model for the rabbit eye [36]. Thus, an injection close to the retina leads to significantly higher drug concentrations in the posterior segment of the eye than an injection in the center or close to the lens. This aspect could not be imaged with the artificial VS because anatomical components such as the lens cannot be simulated until now. There is also a discernible difference between the fresh and thawed vitreous bodies, suggesting that the structure of the vitreous body is damaged by freezing. The destruction of certain structures during slow freezing processes has also been shown for other tissues such as muscle tissue samples [37,38].

All experiments were performed at room temperature. The vitreous body’s temperature ranges from 34 to 35 °C [39]. Since the temperature could affect the viscosity and thus the diffusion, this aspect should be considered in the future. Further physiological aspects, such as clearance or aqueous flow, are not yet considered here and may influence diffusion. Due to the dry storage, the time during which these experiments can be performed is limited. The storage conditions between measurements would have to be reconsidered to investigate the release from intravitreal implants with longer dissolution times. The porcine vitreous body used in this work was a substitute for the human vitreous. Both have similar structural properties, but further studies with human vitreous bodies are needed to transfer the data to humans. The human vitreous body is, nevertheless, highly variable interindividually and shows different properties depending on the degree of age-related liquefaction. It has to be assumed that the animal eyes used here do not show any liquefaction. It becomes almost impossible to reproduce all conditions of the human vitreous body and to ensure exact transferability. However, it may be possible to find suitable structures for vitreous substitutes and to represent extreme states (high/low degree of liquefaction or vitrectomy). This work aims to present a methodology to identify and compare suitable candidates for these substitutes. Ideally, this could reduce the number of test animals required in preclinical studies for new intravitreal dosage forms. In addition to these ethical considerations, in vitro tests typically also have the advantage of low cost compared to in vivo studies.

In summary, the PAA gel seems to reflect the porcine vitreous body more closely than the HA gel with respect to the observed distribution of the contrast agent. By using MRI, the distribution can be followed continuously and non-destructively. Even though the injected GadoVist^®^ is only a model substance with little relation to therapeutically used agents, a first evaluation and comparison of the investigated VS are possible.

## 5. Conclusions

In vitro VS represent an ethically justifiable and economical method for the preclinical investigation of new dosage forms for intravitreal application. The MRI method presented here offers a possibility to evaluate VS regarding distribution and diffusion processes. Further investigations with other dosage forms, such as suspensions or implants, as well as the use of contrast agents with properties similar to therapeutically applied drugs, may represent further steps in the development of new in vitro VS.

## Figures and Tables

**Figure 1 pharmaceutics-15-00786-f001:**
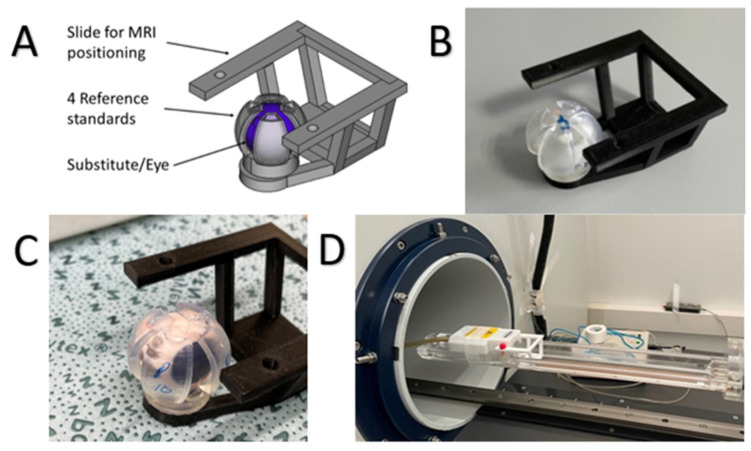
The 3D printed sled for positioning the vitreous substitute/vitreous. CAD model (**A**), 3D printed sled with vitreous substitute (**B**), and porcine eye (**C**). Attached sled in MRI apparatus (**D**).

**Figure 2 pharmaceutics-15-00786-f002:**
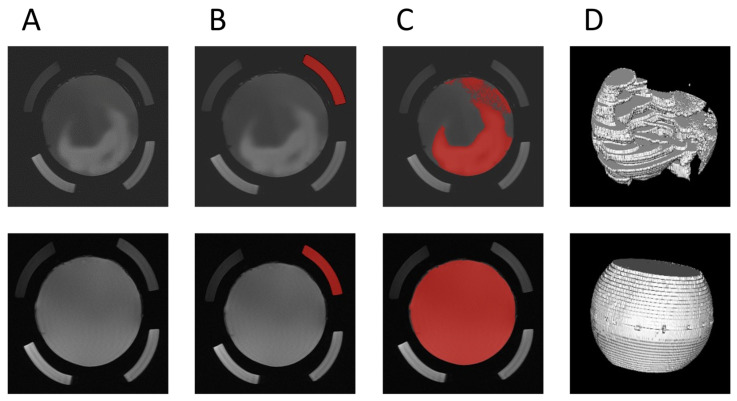
Evaluation of MRI images of artificial vitreous using the example of a PAA gel directly after injection (upper images) and after 12 h (lower images). Contrast agent distribution in the original image (**A**), marking and calculation of the intensity limit (**B**), automatic marking of all intensities above the intensity limit multiplied by 1.25 (**C**), and 3D analysis of the volume spiked with contrast agent (**D**).

**Figure 3 pharmaceutics-15-00786-f003:**
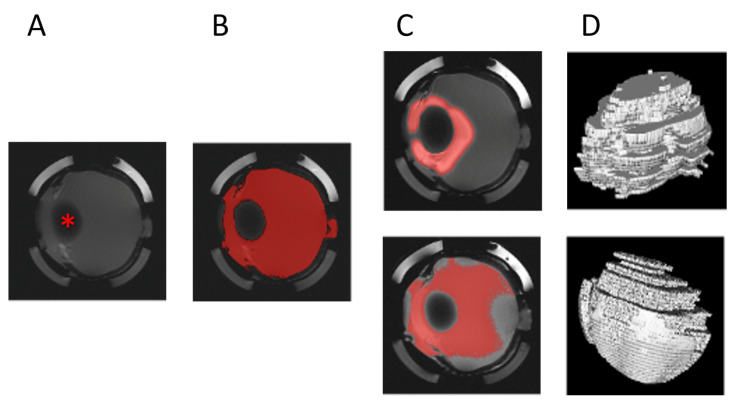
Evaluation of MRI images of porcine vitreous. Image of native vitreous before injection (**A**) and marking and calculation of intensity limit (**B**). Automatic marking of intensities above the intensity limit (multiplied by 1.25) after injection ((**C**) top) and after 12 h ((**C**) bottom). The 3D calculation of the volume spiked with contrast agent after injection ((**D**) top) and after 12 h ((**D)** bottom). Red asterisk marks the lens in the native image.

**Figure 4 pharmaceutics-15-00786-f004:**
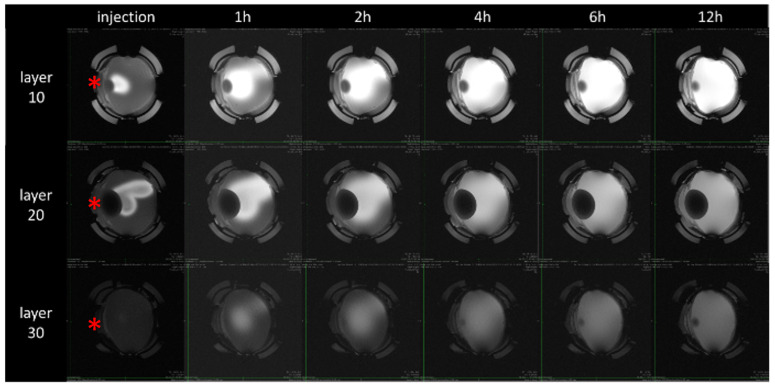
Distribution of the contrast agent GadoVist^®^ over different layers in the fresh ex vivo porcine vitreous body (coronal orientation). Due to the distance from the coil, the lower slices give a darker image. Red asterisk marks the lens.

**Figure 5 pharmaceutics-15-00786-f005:**
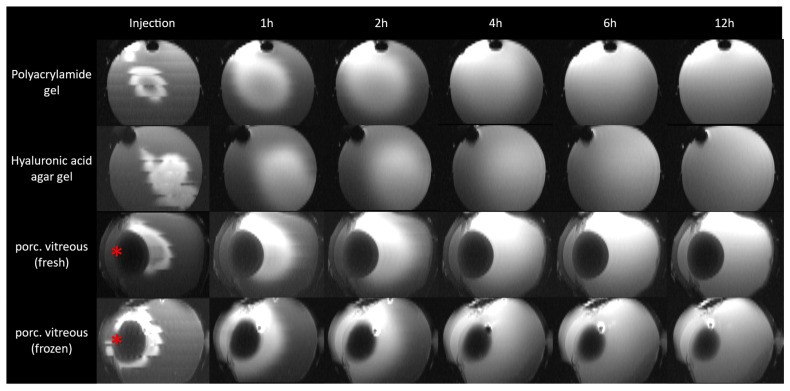
Distribution of the contrast agent GadoVist^®^ over time in different vitreous substitutes and in the ex vivo porcine vitreous body. The coronally measured layers were displayed in a transverse orientation for visualization using HOROS. Red asterisk marks the lens.

**Figure 6 pharmaceutics-15-00786-f006:**
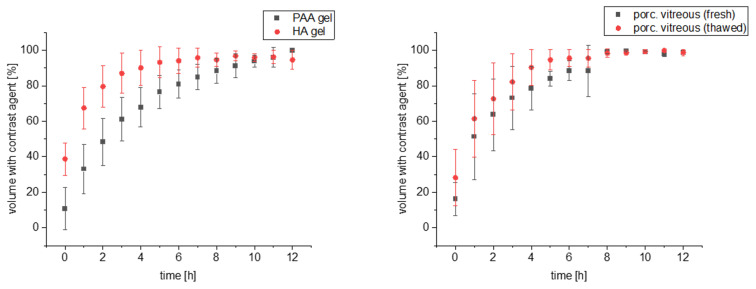
Distribution of the contrast agent GadoVist^®^ within the VS (left) or in the ex vivo porcine vitreous expressed as the volume in which the contrast agent was detected over time. Mean ± SD (n = 3).

**Table 1 pharmaceutics-15-00786-t001:** Composition of the artificial vitreous substitutes. TEMED = Tetramethylethylendiamine, PBS = phosphate-buffered saline, APS = Ammonium persulfate.

Polyacrylamide Gel		Hyaluronic Acid Agar Gel	
Rotiphorese (37.5:1)	6.69 g	Hyaluronic acid	0.15 g
APS-Solution (10%)	1 g	Agar	0.1875 g
TEMED	0.1 g	PBS pH 7.4	Ad 100.0 g
PBS pH 7.4	Ad 100.0 g		

**Table 2 pharmaceutics-15-00786-t002:** MRI sequence parameters.

Parameter	Value
Field of view	40 mm
Slice thickness	0.5 mm
Interslice gap	0 mm
Slices	40
Voxel size	0.089 × 0.089 × 0.5 mm^3^
Repetition time (TR)	845 ms
Echo time (TE)	14 ms
Flip angle	180°
Aquisition time	13:35 min

**Table 3 pharmaceutics-15-00786-t003:** Physicochemical properties of vitreous substitutes (VS) compared with literature values for the human vitreous; values of VS each as MW ± SD (n = 3).

VS	pH Value	Density (g/cm^3^)	Viscosity (mPa·s)
Human vitreous [26,27]	7.4–7.52	1.0053–1.0089	300–2000
Porcine vitreous body	7.42 ± 0.02	1.0041 ± 0.0027	43.1 ± 5.2
Polyacrylamide gel	7.32 ± 0.01	1.0110 ± 0.0009	660 ± 18.1
Hyaluronic acid-agar gel	7.31 ± 0.01	1.0077 ± 0.0000	62.3 ± 3.1

## Data Availability

The data presented in this study are available on request from the corresponding author (T.A.).

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
