# Peer review of "Ex Vivo Visualization of Distribution of Intravitreal Injections in the Porcine Vitreous and Hydrogels Simulating the Vitreous"

_pharmaceutics, 2023, doi:10.3390/pharmaceutics15030786_

Round 1
Reviewer 1 Report
Overall, the authors' paper is good and has scientific validity and novelty in the work.
Well explained. Accepted in present form.
Author Response
Thanks for reading and reviewing our paper.
Reviewer 2 Report
Accept with minor revision.
Currently highlighted the ex vivo model for vitreous drug delivery. Author well justified the problem, Method, result and their interference.
Comments :
1. Here, Eye taken from local slaughter house. Kindly add justification what about live animal and role of nerve system.
2. What is cost in comparison to available model ?
3. How can this data can translate into human?
4. Drawback of magnetic resonance on eye and also challenge for implant drug visualization.
add the about part justification in discussion.
Thanks
Author Response
Thank you for reviewing our paper and the comments attached. In the following, we address the individual points.
- Here, Eye taken from local slaughter house. Kindly add justification what about live animal and role of nerve system
For reference to ex vivo data, the porcine vitreous was selected for comparison with the artificial hydrogels. To avoid non-mandatory animal sacrifice, eyes were used that would be thrown away as by-products when the pigs were slaughtered. The vitreous body has no nervous system running through it, so the discussion did not consider this. One of the main ideas of the use of the gel-based models is to reduce the number of animal studies required. This was further emphasized in the manuscript (line 80-81).
- What is cost in comparison to available model ?
The work presented here demonstrated, to the authors' knowledge, for the first time the possibility of constant tracing of model substances in artificial gels as vitreous substitutes. For this reason, comparison of costs with existing models is difficult. They are significantly lower if the costs are compared with those of animal studies. Compared to other in vitro experiments, the only relevant cost factor is the use of MRI, which is the only way to achieve the advantage of the continuous setup. A comment on cost was added in the discussions section (line 384 – 386).
- How can this data can translate into human?
The reviewers have addressed a good point and certainly the greatest unknown of this work. The human vitreous is highly variable between individuals and has different properties depending on the degree of age-related liquefaction. It becomes nearly impossible to represent all states of the human vitreous body and to guarantee exact transferability. However, finding suitable structures for vitreous substitutes and representing extreme states (high/low degree of liquefaction/vitrectomy) is possible. This work presents a methodology to identify and compare suitable candidates for these substitutes. It would certainly be interesting to study human vitreous as a comparison for a transfer to humans. However, there is again the problem of high variability, which is challenging to implement for a robust in vitro method. In addition, other physiological aspects still need to be considered in vitro. Regarding the commentary, a critical consideration of transferability was added to the manuscript (lines 374-381).
- Drawback of magnetic resonance on eye and also challenge for implant drug visualization
The influence of MRI on the eye is relatively small for the short examination period. A loss of fluid could be problematic, which, depending on the extent, could lead to a deformation of the eye and, thus, to a change in the volumes. This could not be observed for the selected investigation period of 12 hours. The storage conditions between the individual measurements must be reconsidered to follow more extended releases or distribution processes. Also, the more extended release could require a high loading by contrast agent, which could distort the signal intensities. Finally, only a distribution can be visualized with this method. Thus, modeling of the clearance would be necessary to examine intravitreal implants, which is currently not implemented. A paragraph regarding the examination time was added to the manuscript (line 371-374)

Reviewer 3 Report
General comments
This paper describes the use of MRI to image the diffusion of a contrast agent, Gado Vist-R in polyacrylamide gels, agar-hyaluronate (HA) gels and porcine eyes after intragel and intravitreal injection over a period of 12 hours. MRI imaging was used in order to evaluate material diffusion without disrupting the gels by sampling. The authors state that in vitro studies on intravitreal pharmacodynamics has been insufficiently studied. In so far as this is a study of intragel diffusion the experiment is well performed and the data robust. There are some novel methods used here to evaluate diffusion in the gels. it is not clear whether this is a study of optimal vitreous substitutes or of diffusion of materials with a modelled vitreous; on balance it appears to be a study of modelling of conditions approximating a "normal vitreous gel" which could be used to study how materials behave when injected into the vitreous cavity. This could be better emphasised.
While the study is interesting, they do not justify why such studies might have advantages over in vivo studies. In particular, the role of the well known antero-posterior aqueous fluid flow and consequent vitreous currents may impinge on concentration-dependent diffusion within the vitreous. Therefore how directly applicable this study is to intravitreal injections in humans is unclear.
Specific comments
· The final concentration / percentage of the PAA and agar-HA gels hyaluronate should be documented and their justification as vitreous substitutes validated.
· The molecular weight of HA greatly modifies diffusion rate. The HA MW is not specified.
· Diffusion rate varies with temperature. In vivo temperature is 37 degrees C; the experiments performed here were at room temperature.
· reference 28 is incomplete
· on line 344 what is meant by "injection sport" ?
Author Response
Thank you for reviewing our paper and the comments attached. In the following, we address the individual points.
General Comments:
- „it is not clear whether this is a study of optimal vitreous substitutes or of diffusion of materials with a modelled vitreous; on balance it appears to be a study of modelling of conditions approximating a "normal vitreous gel" which could be used to study how materials behave when injected into the vitreous cavity. This could be better emphasised.“
We are sorry that the intention of this work was not clear. The aim was to develop a method to compare artificial vitreous substitutes for distribution processes. In developing new in vitro substitutes, this method could characterize potential candidates and compare distribution patterns with those ex vivo. For clarification, additional lines were added to the introduction (lines 80-81) and discussion (lines 381-384).
- „While the study is interesting, they do not justify why such studies might have advantages over in vivo studies. In particular, the role of the well known antero-posterior aqueous fluid flow and consequent vitreous currents may impinge on concentration-dependent diffusion within the vitreous. Therefore how directly applicable this study is to intravitreal injections in humans is unclear.“
The reviewers have addressed a good point and certainly the greatest unknown of this work. The human vitreous is highly variable between individuals and has different properties depending on the degree of age-related liquefaction. It becomes nearly impossible to represent all states of the human vitreous body and to guarantee exact transferability. However, finding suitable structures for vitreous substitutes and representing extreme states (high/low degree of liquefaction/vitrectomy) is possible. This work presents a methodology to identify and compare suitable candidates for these substitutes. Ideally, this will reduce the number of test animals required in preclinical studies, which will be a benefit. It would be interesting to study human vitreous as a comparison for transferring to humans. However, there is again the problem of high variability, which is challenging to implement for a robust in vitro method. In addition, other physiological aspects mentioned by the reviewers still need to be considered in vitro. Regarding the commentary, a critical consideration of transferability was added to the manuscript (lines 372-381).
Specific Comments:
- The final concentration / percentage of the PAA and agar-HA gels hyaluronate should be documented and their justification as vitreous substitutes validated.
Concentration of the hydrogels were added to the manuscript (line 91 and 97).
- The molecular weight of HA greatly modifies diffusion rate. The HA MW is not specified.
Molecular weight of HA was added to the manuscript (line 98).
- Diffusion rate varies with temperature. In vivo temperature is 37 degrees C; the experiments performed here were at room temperature
The reviewers have addressed an important point here. Temperature affects diffusion on the one hand, and vitreous substitutes on the other. Non-published data from previous studies show at least no significant changes in the hydrogels between 25 °C and 35 °C. This point was added to the discussion (lines 367-370).
- reference 28 is incomplete
Rerefence 28 was modified and should be correct now
- on line 344 what is meant by "injection sport" ?
This was a missspelling, thanks for the comment (line 346 sport changed to spot).
